# A Fatal Case of Metastatic Pulmonary Calcification during the Puerperium

**DOI:** 10.3390/ijms232315131

**Published:** 2022-12-01

**Authors:** Alberto Chighine, Andrea Corona, Gualtiero Catani, Celeste Conte, Roberto Demontis, Matteo Nioi

**Affiliations:** 1Department of Medical Sciences and Public Health, Section of Legal Medicine, University of Cagliari, 09042 Monserrato, Italy; 2Kantonsspital Graubünden, 7000 Chur, Switzerland; 3Department of Health Surveillance and Bioethics, Università Cattolica del Sacro Cuore, Fondazione Policlinico Universitario Agostino Gemelli IRCCS, 00168 Rome, Italy

**Keywords:** primary hyperparathyroidism, forensic, pulmonary metastatic calcification

## Abstract

We present an unusual case of a fatal respiratory failure in a young woman developed two weeks after she gave birth at home. Circumstantial and clinical features of the case were strongly suggestive for a ‘classical’ septic origin of the respiratory symptoms. Autopsy, together with histopathological and immunohistochemical analyses allowed demonstrating a massive calcium redistribution consisting of an important osteolysis, especially from cranial bones and abnormal accumulation in lungs and other organs. Such physiopathology was driven by a primary hyperparathyroidism secondary to a parathyroid carcinoma as demonstrated by immunohistochemistry. This very rare case is furthermore characterised by a regular pregnancy course, ended with the birth of a healthy new-born. A complex interaction between pregnancy physiology and hyperparathyroidism might be hypothesised, determining the discrepancy between the relative long period of wellness and the tumultuous cascade occurred in the puerperium.

## 1. Introduction

Primary hyperparathyroidism (pHPT) is one of the most common endocrine disorders in the adult population, being significantly higher in women. A solitary parathyroid adenoma (80%) generally sustains pHPT, whereas only a residual number of cases (1%) are caused by malignancies [1]. Hyperparathyroidism results in high parathormone and serum calcium levels, the latter being responsible for symptoms classically addressed as ‘*bones, stones, abdominal moans, and psychic groans*’, consistent in fractures, nephrolithiasis, gastritis, and psychiatric symptoms. Although rare, hypertension, rhythm disorders, and pancreatitis may occur as well [2].

This disorder rarely strikes during pregnancy and several issues make the diagnosis in pregnant women extremely challenging. *Hyperemesis gravidarum* may indeed mask hypercalcemia symptoms while calcium levels are seldom elevated because of pregnancy physiological modifications (haemodilution, hypoalbuminemia, maternal hypercalciuria and high foetal calcium demand, amongst others) [3]. In pregnancy, symptoms may even be more occult with the possibility of severe events such as pre-eclampsia and hypercalcaemic crisis, not to mention foetal complications [4].

What is more, in cases of women asymptomatic or with unspecific symptoms, calcium levels are not routinely screened during pregnancy [5]. Once diagnosed, as pHPT presents significant complication rates on both mother and foetus, surgical approach is the treatment of choice, although it should be ideally performed during the second trimester. The risk of complications seems even higher after delivery, as pregnancy has a ‘buffer’ effect on hypercalcemia cessation [6].

Only a few reports have described a metastatic pulmonary calcification (MPC) secondary to pHPT [7,8,9].

We present the case of a 35-year-old woman, experiencing severe symptoms during the second week of puerperium following regular pregnancy and childbirth. To the best of the authors’ knowledge, this is the first report of a pHPT occurring in the puerperium complicated by a severe MPC, which eventually led to a fatal respiratory failure.

## 2. Case Report

### 2.1. Medical History and CT Findings

We present the case of a 35-year-old tertigravida (160 cm × 55 kg) with past medical history positive for aspirin allergy; no endocrine disorders were revealed in the familiar anamnesis. The patient underwent routine check-ups during pregnancy showing slight anaemia, whereas electrolytes were never investigated. Echography check-ups ruled out foetal abnormalities. A home-delivery was planned and conducted by professional midwives. Gestation and delivery occurred without clinical complications and mother and new-born underwent routine gynaecologist and paediatric postpartum home check-ups.

After the first week of puerperium, the patient experienced low back and leg pain, responsive to painkillers. During the second puerperal week a worsening tachypnoea settled in, shifting in severe dyspnoea on the 14th day postpartum. On the next day the patient was admitted to the emergency department, showing clinical signs of shock (blood pressure = 100/60 mmHg, heart rate = 130 bpm, high respiratory frequency with face mask 10 L/min and SpO_2_ = 80%, weak peripheral pulses, cold and mottled limbs) and decreased breath sounds on both sides. Blood and arterial gas tests at admission are reported in Appendix A (Appendix A).

Arterial blood gas test (Appendix A) was consistent with a primary hypoxemic respiratory insufficiency and metabolic acidosis with inadequate respiratory compensation. A contrast CT scan ruled out pulmonary embolism and cardiac tamponade whereas provided evidence of bilateral diffused parenchymal consolidation was consistent with an inflammatory aetiology. A 20 mm inhomogeneous thyroid nodule was also spotted on the left lobe.

Because of respiratory and haemodynamic instability, the patient underwent a mechanical ventilation trial and was admitted to the intensive care unit (ICU) with the suspicious of pneumonia with septic shock and multi organ failure (MOF). The patient was conscious but anxious, objective examination highlighted hypothermia (34.7 °C), tachypnoea (respiratory rate > 25/min), severe bilateral reduction of breath sounds, eyes and mouth dryness, diffused skin mottling, and perioral cyanosis. A wide spectrum antibiotic therapy was initiated, together with fluid supplementation and inotropic support. Soon after admission she was intubated and ventilated with O_2_ 100%; arterial gas test parameters at ICU admission are shown in Appendix A.

Less than 2 h after ICU admission the patient experienced a cardiac arrest, successfully treated with resuscitation manoeuvrers. After return of spontaneous circulation, a chest X-ray showed a massive consolidation on both lungs’ parenchyma. A large amount of secretion was aspirated from endotracheal tube with mixed content of enteric fluid and pulmonary oedema. A second cardiac arrest did not answer to resuscitation manoeuvres and, six hours after the emergency department admission, the patient died.

### 2.2. Autopsy Findings

A judicial autopsy was ordered by the local prosecutor’s office. A team of forensic and clinical pathologists performed the autopsy. Organ weights are reported in Table 1.

Most remarkable macroscopic findings are reported as follows. Skullcap (Figure 1) appeared diffusely eroded in a grainy reddish pattern resulting in a global bone thinning more accentuated in temporal and parietal regions.

The right thyroid lobe was enlarged due to a 2.5 cm diameter mass, compact and greyish on section. Lungs (Figure 2a) were utterly increased in weight, hyper-inflated, and consolidated. Diffused subpleural petechiae were present bilaterally. On section (Figure 2b), parenchyma appeared diffusely grey pinkish, extremely compact with porous appearance, and, when squeezed, gave a crunchy sensation and released abundant, slightly foamy, liquid.

Internal genital findings were consistent with puerperium. Due to personal beliefs of the family, placenta was unconventionally preserved (under salt) while macroscopic evaluation did not highlight any noticeable pathological finding.

### 2.3. Histopathological and Immunohistochemical Findings

Histopathological analysis was performed on all the specimens collected. Skullcap bone (Figure 3) showed a diffused demineralization with multiple foci of hyperplastic multinucleate osteoclasts and osteofibrosis.

Thyroid mass detected at CT scan was better identified as a parathyroid carcinoma consisting of a capsulated lesion with proliferation of round cells with slight nuclear atypia, anisokaryosis, and scarce cytoplasm, infiltrating the capsule and the surrounding thyroid parenchyma (Figure 4a). Immunohistochemical analyses of such lesion revealed positivity for CK8/CK18, chromogranin (Figure 4b), and NSE, and negativity for CK19, CD31, calcitonin and thyroglobulin. Cellular proliferation index (Ki67) was <1%.

Lungs were characterised by a massive, diffuse calcium deposition regarding alveolar septa and bronchi walls (especially in basal membrane and submucosa). Analogous deposits were found in arterial and venous walls. Alveoli were filled with amorphous material, slightly eosinophils, mixed with red blood cells and epithelial cells (Figure 5a,b). Some sections provided evidence of neutrophile exudation in both bronchi lumen and alveoli. A diffuse calcium phosphate deposition was confirmed with von Kossa stain (Figure 5c).

A diffuse nephrocalcinosis was bilaterally spotted; renal arterial walls were also involved by calcium deposition (Figure 6b). Diffuse calcifications were also observed in the gastroenteric tract, particularly in the gastric mucosa lamina propria and small vessels’ walls (Figure 6a).

Taken together, histopathological and immunohistochemical findings were consistent with a solid-trabecular parathyroid carcinoma and a widespread calcium deposition, particularly severe on lungs, kidneys, and gastric wall. Alveolar septa, bronchi, and vessels walls were massively affected by calcium deposition; lungs also presented multiple foci of acute alveolar pneumonia.

To the best of the authors’ knowledge, the new-born was reported in good health and, to date, has never experienced any complication.

## 3. Discussion

The case presented here was a harsh challenge for clinicians due to the rapid onset of symptoms that led to death within just a few hours after the admission into hospital. The difficult differential diagnosis reflected a peculiar physiopathology based on the complex interaction of a rare pathology and pregnancy.

The patient died before a defined diagnosis was made. Clinical course, CT scan, lab parameters, and circumstances (14 days following home delivery) were highly suggestive of a severe septic shock with respiratory failure resulting in MOF. On the other hand, the clinical course did not allow confirmation, nor excluding, of an infection.

Autopsy and histopathological analyses were essential to unravel the clinical conundrum. Post-mortem investigations ascertained the aetiology of the respiratory failure and its physio-pathological basis. Firstly, the thyroid mass described on CT scan was found instead to be a parathyroid carcinoma. Secondly, autopsy provided evidence of a massive calcium redistribution in the body due to the gross finding in the skullcap and the microscopic analyses of lungs and, to a lesser extent, kidneys, and stomach.

Combined with clinical biochemistry parameters, these findings indicate a condition of pHPT. Clinical suspicion was diverted from HPT since parameters’ modification could have supported terminal pneumonia. With the benefit of the hindsight, others may be owing to possible paraneoplastic activity.

A few cases of respiratory symptoms sustained by calcium deposition were reported in literature, often linked to pHPT due to parathyroid adenoma [7,8,9], none of which regarded pregnant women. In other reports, MPC was linked to another medical conditions, such as multiple myeloma [10] and chronic renal failure [11].

Macroscopic, histopathological and immunohistochemical lungs findings suggested an etiological relationship between MPC and the developing of respiratory failure. Massive calcium deposition can indeed alter, thickening, the alveolar capillary membrane leading to an alteration of the oxygen and carbon dioxide exchange.

It is hard to exclude the influence of the recent pregnancy in such peculiar physiopathology. Although the understanding of the complex interaction between these biological phenomena is way beyond that of the aim of this report, a possible explanation may be addressed.

The parathyroid carcinoma was most likely already present during the pregnancy, which happened without clinical complications or specific symptoms. Physiological changes linked to pregnancy, such as haemodilution, hypoalbuminemia, maternal hypercalciuria, and high foetal calcium demand may have buffered the hypercalcemia, preventing the symptoms. We are not able to prove the occurrence of calcium redistribution yet during pregnancy, since no specific screenings were conducted. However, this hypothesis may be supported in its entirety due to the massive erosions spotted on the skullcap. The resulting physio-pathological and clinical stability may have dramatically ended in the puerperium as the ‘buffer’ mechanism ceased, with the unspecific symptoms’ onset at the end of the first week and the tragical course on the 14th day with dyspnoea rapidly evolving into a massive, fatal respiratory insufficiency.

The case presented here has several limitations. As pregnancy was reported to occur without complications, only routine screenings were performed, regarding the calcium/phosphate balance. Due to unconventional preservation of placenta, pathological findings may have been lost. The CT report without images was only available to the authors.

## 4. Conclusions

The case presented provides evidence of lung complications in the case of a pHPT, with a silent clinical course during pregnancy, resulting in a fatal MPC occurred in the puerperium. This report hence suggests a potential implementation of calcium metabolism screening during pregnancy even when specific symptoms are missing. The authors believe that sharing this unique case with the scientific audience stresses the importance of the clinical autopsy either in completing a diagnosis or preventing potential litigations.

## Figures and Tables

**Figure 1 ijms-23-15131-f001:**
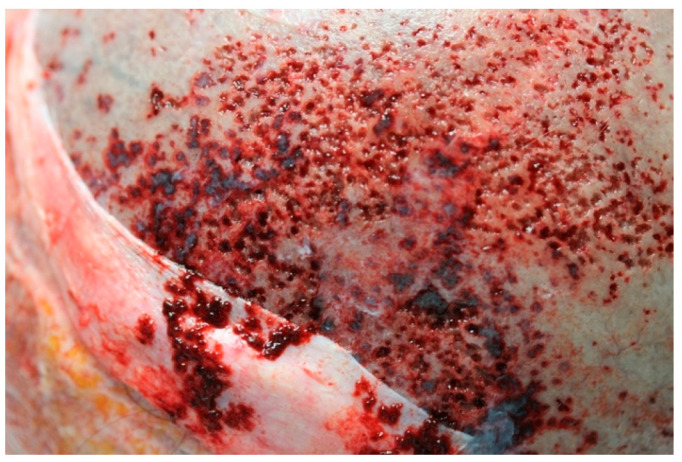
Skullcap erosion, parietal bone. Macroscopic finding.

**Figure 2 ijms-23-15131-f002:**
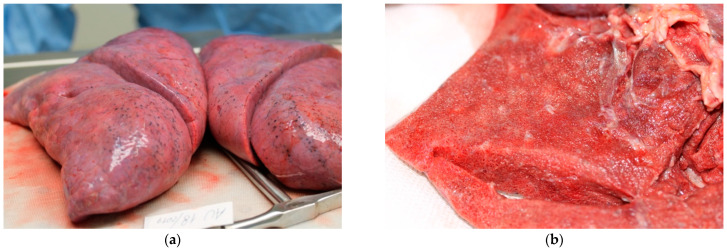
Lungs, macroscopic findings. (**a**) Hyper-inflated and consolidated lungs with thick margins; (**b**) on section, pulmonary tissue appears compact and porous.

**Figure 3 ijms-23-15131-f003:**
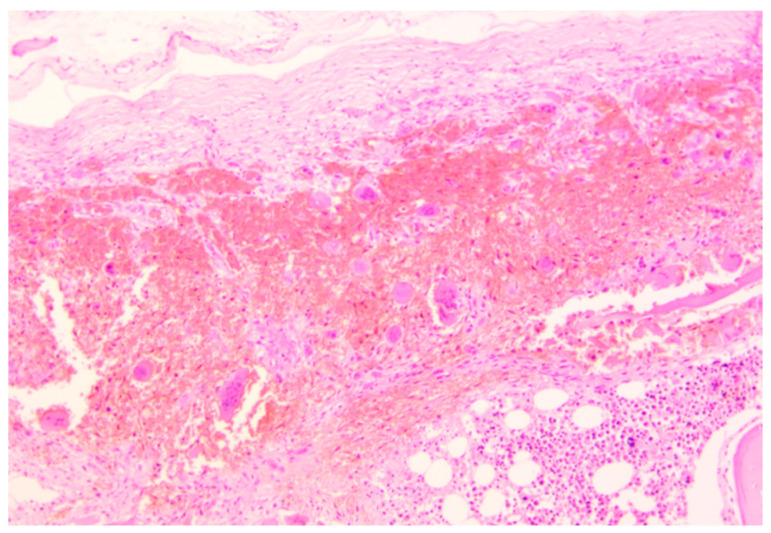
Skullcap erosion, parietal bone. H&E staining—magnification 40×. Immediately below the external theca of the bone cap is present a large osteolytic hotbed, bone tissue with massive demineralization replaced by granulation tissue, and massive red blood cell infiltration.

**Figure 4 ijms-23-15131-f004:**
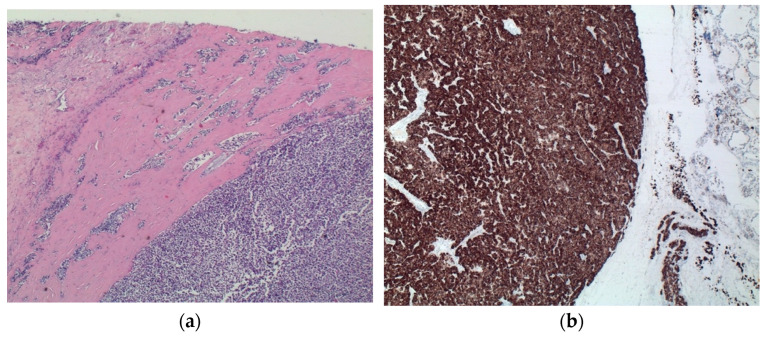
Parathyroid. (**a**) H&E staining—magnification 120×. The nodule tightly stuck to the thyroid disclosed a uniform patternless solid proliferation of round cells with a scant amount of cytoplasm, seldom arranged in a lobular pattern, surrounded by a dense fibrous capsule that adheres to adjacent tissues. Nuclei are irregular and hyperchromic. There is evidence of capsular and vascular invasion. (**b**) IHC chromogranin—magnification 120×. Cells are widely positive compared to negative thyroid tissue (right side).

**Figure 5 ijms-23-15131-f005:**
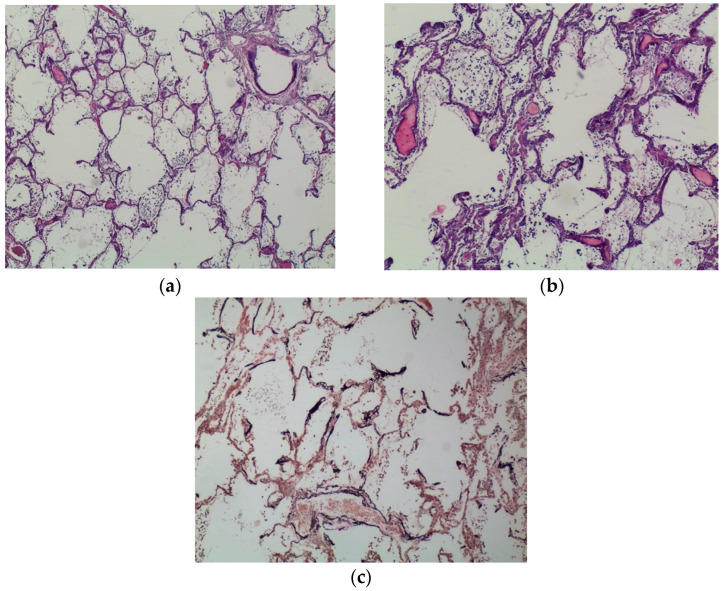
Lungs. (**a**) H&E staining—magnification 120×. Alveolar pattern and bronchial architecture are preserved although the septal walls are slightly thickened and basophilic. Calcium deposition can be spotted in a small bronchus basal membrane. (**b**) H&E staining—magnification 240×. The walls of small blood vessels and alveoli are particularly affected by metastatic calcification with a heavy diffuse deposition of calcium salts resulting in thickened, basophilic, and vaguely hyaline alveolar walls. The alveolar cavities contain few leukocytes and scarce amount of amorphous material. (**c**) IHC von Kossa stain—magnification 120×. Calcium salts deposition appears as dark stain because of its affinity for silver nitrate.

**Figure 6 ijms-23-15131-f006:**
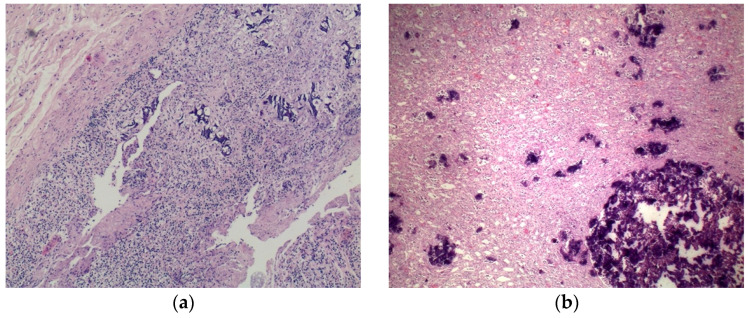
Metastatic calcification involving viscera—calcium is recognisable in routine sections due to its affinity for haematoxylin. (**a**) Stomach, H&E staining—magnification 60×. Granular depositions of amorphous calcium salts in the lamina propria of the gastric mucosa. (**b**) Kidney, H&E staining—magnification 60×. Granular depositions of amorphous calcium salts in the kidney medullar interstitium.

**Table 1 ijms-23-15131-t001:** Organ weights in grams (g).

Brain	1300
Heart	300
Right lung	1430
Left lung	1300
Liver	850
Spleen	250
Right Kidney	200
Left Kidney	180

## Data Availability

Data, after adequate anonymization, will be available on request from the corresponding author (A.C.).

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
