# Peer review of "A Fatal Case of Metastatic Pulmonary Calcification during the Puerperium"

_ijms, 2022, doi:10.3390/ijms232315131_

Round 1

Reviewer 1 Report

The authors describe a case that appears to have no precedent in the published literature, of a woman developing extensive pulmonary calcification soon after delivery of an infant. I have attempted my own quick and dirty literature search, and I agree with their view that there are no other comparable cases. However, I do not think it appropriate to designate this as a case of Adult Respiratory Distress Syndrome (ARDS), because ARDS is characterised by progressive difficulty in maintaining oxygenation of the blood, with a typical chest x-ray appearance: neither of these features have been demonstrated in this case (although I note that Line 87 promises some biochemical parameters in “table 2”, but there is no Table 2 in the manuscript). Further, almost all cases of ARDS investigated at autopsy show diffuse alveolar damage (DAD), as indicated in the brief review by Yoshikawa and Bychkov (2022). DAD is characterised microscopically by “hyaline” membranes of acellular proteinaceous material lining the alveolar spaces, similar to those seen in fatal cases of coronavirus disease-19 (Ramadori, 2022); such membranes are very clearly absent from the three images in Figure 4. I suggest that the title should be changed to “A fatal case of metastatic pulmonary calcification during the puerperium”: not all acute respiratory distress is an example of the acute respiratory distress syndrome. Other references to ARDS could also be removed, without any detraction from the central message of the paper

I have, on very few occasions, seen similar calcification in lung tissue at autopsy, but these are always been associated with hyperparathyroidism in associated with chronic renal failure and dialysis; in each of two cases I have dealt with personally, dialysis has been complicated by poor compliance from the patient.

Lines 54 to 56 appear to be a direct quotation from the Instructions to Authors, and I suspect an error in cutting and pasting.

Line 67: The term “tachy-dyspnoea” is not normally used in English language medical literature

Line 69: The abbreviation “HF” is not standard

Figure 1b: This microscopic image requires improvement, particularly because there are several particles of dirt on the surface of the slide. A new photograph should be taken after cleaning the slide, or if necessary, after producing a new section.

Line 200: In my experience, metastatic pulmonary calcification is most often seen in association with chronic renal failure. Georges and Srour (2016) describe one case, but there are several more in the literature.

Lines 204-5: The histological sections provide no evidence for diffuse alveolar damage. The

Georges S, Srour N. Metastatic pulmonary calcification in end-stage renal failure. CMAJ. 2016 Oct 18;188(15):E394. doi: 10.1503/cmaj.150778. Epub 2016 Aug 2. PMID: 27486208; PMCID: PMC5056891.

Ramadori GP. SARS-CoV-2-Infection (COVID-19): Clinical Course, Viral Acute Respiratory Distress Syndrome (ARDS) and Cause(s) of Death. Medical Sciences. 2022; 10(4):58. https://doi.org/10.3390/medsci10040058

Yoshikawa A, Bychkov A. ARDS / DAD. PathologyOutlines.com website. https://www.pathologyoutlines.com/topic/lungnontumordiffusealveolardamage.html. Accessed November 11th, 2022.

Author Response

Dear Reviewer,

We are sincerely grateful for your comprehensive and insightful comments. Kindly be informed that we have made the relevant corrections and modifications and we are now resubmitting the revised manuscript for further consideration. Please also note that manuscript has been extensively revised by an English native-speaker.

We would greatly appreciate if our paper could be favourably considered for publication in IJMS in the current revised version.

Number-wise answers to specific comments/suggestions are as follows:

The authors describe a case that appears to have no precedent in the published literature, of a woman developing extensive pulmonary calcification soon after delivery of an infant. I have attempted my own quick and dirty literature search, and I agree with their view that there are no other comparable cases.

Comment 1. However, I do not think it appropriate to designate this as a case of Adult Respiratory Distress Syndrome (ARDS), because ARDS is characterised by progressive difficulty in maintaining oxygenation of the blood, with a typical chest x-ray appearance: neither of these features have been demonstrated in this case (although I note that Line 87 promises some biochemical parameters in “table 2”, but there is no Table 2 in the manuscript). Further, almost all cases of ARDS investigated at autopsy show diffuse alveolar damage (DAD), as indicated in the brief review by Yoshikawa and Bychkov (2022). DAD is characterised microscopically by “hyaline” membranes of acellular proteinaceous material lining the alveolar spaces, similar to those seen in fatal cases of coronavirus disease-19 (Ramadori, 2022); such membranes are very clearly absent from the three images in Figure 4. I suggest that the title should be changed to “A fatal case of metastatic pulmonary calcification during the puerperium”: not all acute respiratory distress is an example of the acute respiratory distress syndrome. Other references to ARDS could also be removed, without any detraction from the central message of the paper.
We are very thankful to the reviewer for addressing this issue. Firstly, we must highlight that although it was specifically asked to the Prosecutor Office, we did not have the images of the mentioned CT exam that would have supported the ARDS clinical diagnosis (this issue has been added to limitations). Secondly, the arterial gas test we referred to was mistakenly addressed as table 2 instead of the correct table S1b; such issue has also been addressed.
Lastly, we do agree that DAD was not pronounced in the lungs’ section showed with particular regard to hyaline membrane absence. On the other hand, consistently with Yoshikawa and Bychkov ‘
Not all ARDS show DAD pattern and hyaline membranes might be not present in the first week of ARDS (Thille AW et al. (2013). Chronology of histological lesions in acute respiratory distress syndrome with diffuse alveolar damage: a prospective cohort study of clinical autopsies. The Lancet. Respiratory medicine, 1(5), 395–401. https://doi.org/10.1016/S2213-2600(13)70053-5). We would emphasise that our case suggests a different ARDS pathogenesis than the ‘classical’ infective ones, which should be considered secondary to the thickening of alveolar-capillary membranes due to calcium deposition resulting in alteration of the O2/CO2 exchanges.
All things considered, we do agree on the final comment of the reviewer so we switched the title and the main text contents accordingly, since not all the evidence fully supported the ARDS. We are now referring to a more unspecific respiratory distress, which appears consistent to the arterial gas test and clinical evolution.

Comment 2. I have, on very few occasions, seen similar calcification in lung tissue at autopsy, but these are always been associated with hyperparathyroidism in associated with chronic renal failure and dialysis; in each of two cases I have dealt with personally, dialysis has been complicated by poor compliance from the patient.
We thank the reviewer for sharing her/his remarkable experience regarding MPC. As per Comment 7., MPC during chronic renal failure has been mentioned in the discussion section and reference from George and Srour has been added [11].

Comment 3. Lines 54 to 56 appear to be a direct quotation from the Instructions to Authors, and I suspect an error in cutting and pasting.
Addressed accordingly.

Comment 4. Line 67: The term “tachy-dyspnoea” is not normally used in English language medical literature.
Addressed, the term has been changed to “tachypnoea”.

Comment 5. Line 69: The abbreviation “HF” is not standard.
Addressed, vital parameters have been renamed accordingly.

Comment 6. Figure 1b: This microscopic image requires improvement, particularly because there are several particles of dirt on the surface of the slide. A new photograph should be taken after cleaning the slide, or if necessary, after producing a new section.
We thank the reviewer for the precious suggestion. Unfortunately, the slide was of poor quality, so it was not possible to take a better snapshot from the same one. We hence decide to replace with a different slide, still significant with respect to the pathological condition.

Comment 7. Line 200: In my experience, metastatic pulmonary calcification is most often seen in association with chronic renal failure. Georges and Srour (2016) describe one case, but there are several more in the literature.
The suggested paper has been cited in the discussion [11].

Comment 8. Lines 204-5: The histological sections provide no evidence for diffuse alveolar damage. The
Unfortunately, this comment appears incomplete. We nevertheless hypothesise a similar content to comment 1.

Reviewer 2 Report

The article deals with a case report of a woman who died after childbirth due to parathyroid carcinoma. The case report is well described however there are revisions to be made:

1) The abstract is not very specific, the authors should better specify the results of the autopsy and histology.

2) The caption of figure 2 is wrong, the authors should correct it.

3) Figure 1b should be moved to the histology section as a separate figure.

4) Paragraph of histology: how do the authors explain that the lung metastases are from parathyroid carcinoma? If so, the same positivity/negativity and the same parathyroid markers must be present in the lung, calcification alone is not enough. The same goes for the stomach and kidneys.

5) From line 210 to line 219 the authors maintain that pregnancy has induced a worsening of the pathology due to hormonal changes, however they have not cited any reference. is this their guess? The authors should cite some supporting studies.

However, the case report remains of great interest to the scientific and forensic community.

Author Response

Dear Reviewer,

We are sincerely grateful for your comprehensive and insightful comments. Kindly be informed that we have made the relevant corrections and modifications and we are now resubmitting the revised manuscript for further consideration. Please note that manuscript has been extensively revised by an English native-speaker.

We would greatly appreciate if our paper could be favourably considered for publication in IJMS in the current revised version.

Number-wise answers to specific comments/suggestions are as follows:

The article deals with a case report of a woman who died after childbirth due to parathyroid carcinoma. The case report is well described however there are revisions to be made:

Comment 1. The abstract is not very specific, the authors should better specify the results of the autopsy and histology.
Addressed, abstract has been implemented accordingly.

Comment 2. The caption of figure 2 is wrong, the authors should correct it.
Addressed accordingly.

Comment 3. Figure 1b should be moved to the histology section as a separate figure.
We thank the reviewer for the suggestion. The macroscopic and histopathological pictures were placed next to each other due to the exceptional finding. We hence believe the original intended Figure is captivating to the reader.

Comment 4. Paragraph of histology: how do the authors explain that the lung metastases are from parathyroid carcinoma? If so, the same positivity/negativity and the same parathyroid markers must be present in the lung, calcification alone is not enough. The same goes for the stomach and kidneys.

We thank the reviewer for highlighting this issue. As we reported in the main text (lines 286-288) is only possible to hypothesise a physio-pathological cascade suggesting a calcium redistribution driven by the malignancy activity. Although the word metastases might be misleading, in such case MPC only refers to calcium but not neoplastic cells. It is therefore not possible to ascertain calcium origin with cellular immunohistochemical markers, but only confirm (through Von Kossa Staining) calcium itself.

Comment 5. From line 210 to line 219 the authors maintain that pregnancy has induced a worsening of the pathology due to hormonal changes, however they have not cited any reference. is this their guess? The authors should cite some supporting studies.
Again, it was only possible to give a possible explanation to the findings and we thank the reviewer for giving us the opportunity to clarify it here and, accordingly, in the main text. The genuine idea of the authors was that the massive bone erosion occurred during pregnancy albeit physiological modifications forbid calcium accumulation in lungs, kidney, and stomach. Once the pregnancy has ended the physiological activity, previously buffering the calcium overload, ceased and MPC occurs relatively fast.

However, the case report remains of great interest to the scientific and forensic community.
We thank the reviewer for appreciating our efforts and helping us in improving the manuscript quality.

Round 2

Reviewer 1 Report

The author should be congratulated on the amount of work they have performed, and I think this paper is now much more coherent, at least from a pathological viewpoint. I am very pleased that the authors have replaced the term ARDS, which seems to manage to retain a degree of ambiguity over several decades: I remember arguments (including in the peer-reviewed literature) about whether the “A” stood for Acute or Adult. However, I think the substitute “respiratory distress” may not be understood by all readers, and I think that most, if not all, of the occurrences of “respiratory distress“ in the text would be better rendered as “respiratory failure”.

The table of investigatory results is very useful, but some of the units of measurement are not universal, and it would therefore be helpful for an international readership for a reference range is to be inserted for each measured parameter,

There remain some issues about language, but I think most of these can be sorted at sub-editing. However, I have few specific suggestions for changes that might not be understood by a non-medical language expert:

-          Line 18: I think that “anormal” should be “abnormal“

-          Lines 66-7: It will be surprising to many to read that both gynaecological and paediatric attention was required in the puerperium: in my country that implies a need for visits from specialised medical doctors. Are the authors referring to the “routine” (probably a better word than “regular“ in this context) examinations carried out by specialised nurses?

-          Line 73: A respiratory frequency of 130/minutes seems unrealistic, and I assume that there is a typographic error

-          Line 78: “Passed over” is a lay or journalistic term, and in scientific writing, the word “died” would generally be preferred

-          Lines 112-3: When we write about large lungs at autopsy, the conventional terminology is usually “hyper-inflated”. Increased “consistency” might not be understood: most would simply use the term “firm” or “consolidated”. However, I note that is inconsistent with the later description of a “spongy appearance”. I can appreciate that this is a very difficult thing to describe, but my own recollection of examining lungs with calcification in the context of renal failure was they felt slightly hard, and with a crunching sensation on compression (presumably due to breaking the fragile but calcified intra-alveolar walls).This

-          Line 134: The usual spelling for the thyroid protein is “thyroglobulin”

-          Line 145: I do not understand what is meant by “venous vasa walls”.  Are these the vasa vasorum in the walls of the veins, or is this simply a typographic error?

-          Line 147: “on” should be “in”.

-          Line 148: Strictly speaking, a von Kossa stain demonstrates the presence of phosphate ions, but I think we can avoid being fussy about this. Perhaps the authors might simply write “calcium phosphate”, rather than simply “calcium”

-          Line 162: “Interested” is not quite right; “involved” might be better.

-          Line 175: I suspect “spread” is meant to be “widespread”

-          Line 182: “Impetuous” is not normally used in medical English: “rapid onset of…” might be better.

-          Line 204: “Ex ante” is rarely used in medical English, and the remainder of this sentence might be misunderstood. I suggest something like “clinical suspicion was diverted from HPT…”

I apologise for finding so many other issues for the authors’ attention, but this often happens because major improvements elsewhere make minor idiosyncrasies more obvious. I hope the authors do not find these suggestions oppressive: I think they can be dealt with very quickly.

Author Response

Dear Reviewer,

We are sincerely grateful for your comprehensive and insightful comments. Kindly be informed that we have made the relevant corrections and modifications and we are now resubmitting the revised manuscript.

We would greatly appreciate if our paper could be favourably considered for publication in IJMS in the current revised version.

Number-wise answers to specific comments/suggestions are as follows:

Comment 1. The author should be congratulated on the amount of work they have performed, and I think this paper is now much more coherent, at least from a pathological viewpoint. I am very pleased that the authors have replaced the term ARDS, which seems to manage to retain a degree of ambiguity over several decades: I remember arguments (including in the peer-reviewed literature) about whether the “A” stood for Acute or Adult. However, I think the substitute “respiratory distress” may not be understood by all readers, and I think that most, if not all, of the occurrences of “respiratory distress“ in the text would be better rendered as “respiratory failure”.

We once again thank the reviewer for her/his help in addressing the several issues faced describing a rare clinical case from the forensic and pathology perspective. We do agree that respiratory failure better address the description than respiratory distress and we modified the text accordingly.   

Comment 2. The table of investigatory results is very useful, but some of the units of measurement are not universal, and it would therefore be helpful for an international readership for a reference range is to be inserted for each measured parameter.

Reference ranges for blood test parameters were added, whereas it was not possible reporting arterial gas tests as clinical results did not display any machine-specific reference range.

There remain some issues about language, but I think most of these can be sorted at sub-editing. However, I have few specific suggestions for changes that might not be understood by a non-medical language expert:

-        Comment 3. Line 18: I think that “anormal” should be “abnormal“
Modified accordingly.

-        Comment 4. Lines 66-7: It will be surprising to many to read that both gynaecological and paediatric attention was required in the puerperium: in my country that implies a need for visits from specialised medical doctors. Are the authors referring to the “routine” (probably a better word than “regular“ in this context) examinations carried out by specialised nurses?
We were referring to medical check-ups as well, although in Italy midwives may perform some postpartum check-ups on the mother. The peculiarity of our case is that the mother freely chose to conduct delivery (and puerperium check-ups) at home. We however modified regular with routine.

-        Comment 5. Line 73: A respiratory frequency of 130/minutes seems unrealistic, and I assume that there is a typographic error.
Heart rate was of course intended. Modified accordingly.

-        Comment 6. Line 78: “Passed over” is a lay or journalistic term, and in scientific writing, the word “died” would generally be preferred.
Modified accordingly.

-        Comment 7. Lines 112-3: When we write about large lungs at autopsy, the conventional terminology is usually “hyper-inflated”. Increased “consistency” might not be understood: most would simply use the term “firm” or “consolidated”. However, I note that is inconsistent with the later description of a “spongy appearance”. I can appreciate that this is a very difficult thing to describe, but my own recollection of examining lungs with calcification in the context of renal failure was they felt slightly hard, and with a crunching sensation on compression (presumably due to breaking the fragile but calcified intra-alveolar walls).This
Unfortunately the reviewer’ comment seems to be incomplete. However, we appreciate how the reviewer understand the authors difficulty in describing an unusual and abnormal pulmonary pathological finding in a way that can be divulged towards the whole scientific community. The precious suggestions have been used to modify main text and figures’ legend.

-        Comment 8. Line 134: The usual spelling for the thyroid protein is “thyroglobulin”
Modified accordingly.

-        Comment 9. Line 145: I do not understand what is meant by “venous vasa walls”.  Are these the vasa vasorum in the walls of the veins, or is this simply a typographic error?
We were referring to venous walls. We hence eliminated the word vasa for the sake of clarity.

-        Comment 10. Line 147: “on” should be “in”.
Modified accordingly.

-        Comment 11.  Line 148: Strictly speaking, a von Kossa stain demonstrates the presence of phosphate ions, but I think we can avoid being fussy about this. Perhaps the authors might simply write “calcium phosphate”, rather than simply “calcium”
We do agree. Modified accordingly.

-         Comment 12.  Line 162: “Interested” is not quite right; “involved” might be better.
Modified accordingly.

-        Comment 13.  Line 175: I suspect “spread” is meant to be “widespread”
Modified accordingly.

-         Comment 14.  Line 182: “Impetuous” is not normally used in medical English: “rapid onset of…” might be better.
Modified accordingly.

-        Comment 15.  Line 204: “Ex ante” is rarely used in medical English, and the remainder of this sentence might be misunderstood. I suggest something like “clinical suspicion was diverted from HPT…”
Modified accordingly.

I apologise for finding so many other issues for the authors’ attention, but this often happens because major improvements elsewhere make minor idiosyncrasies more obvious. I hope the authors do not find these suggestions oppressive: I think they can be dealt with very quickly.
We are extremely thankful for the reviewers’ suggestions, and we strongly support critical revision as a constructive way to improve scientific quality.   

Reviewer 2 Report

The manuscript has been sufficiently improved. I recommend moving fig 1b to paragraph 2.3

Author Response

Dear Reviewer,

We are sincerely grateful for your comprehensive and insightful comments. Kindly be informed that we have made the relevant corrections and modifications and we are now resubmitting the revised manuscript.

We would greatly appreciate if our paper could be favourably considered for publication in IJMS in the current revised version.

Number-wise answers to specific comments/suggestions are as follows:

Comment 1. The manuscript has been sufficiently improved. I recommend moving fig 1b to paragraph 2.3

We thank the reviewer for her/his precious comments. Figure 1b was moved to histopathological and immunohistochemical findings.

Round 3

Reviewer 1 Report

I apologise for the apparently incomplete Comment 7: this is in fact an artefact of a digital dictation system that adds the word "this" in response  to the most imperceptible sniff or sigh!